# Changes in Caprine Milk Fat Globule Membrane Proteins after Heat Treatment Using a Label-Free Proteomics Technique

**DOI:** 10.3390/foods11172705

**Published:** 2022-09-05

**Authors:** Daomin Yan, Lina Zhang, Yixuan Zhu, Mengyu Han, Yancong Wang, Jun Tang, Peng Zhou

**Affiliations:** State Key Laboratory of Food Science & Technology, Jiangnan University, Wuxi 214122, China

**Keywords:** milk fat globule membrane proteome, heat treatment, ultra-pasteurized, ultra-high-temperature instant sterilization, spray-dried

## Abstract

Milk proteins are prone to changes during the heat treatment process. Here, we aimed to study the changes in caprine milk fat globule membrane (MFGM) proteins with three heat treatment processes—ultra-pasteurization (85 °C, 30 min), ultra-high-temperature instant sterilization (135 °C, 5 s), and spray-drying (inlet, 160 °C and outlet, 80 °C)—using the label-free proteomics technique. A total of 1015, 637, 508, and 738 proteins were identified in the raw milk, ultra-pasteurized milk, ultra-high-temperature instant sterilized milk, and spray-dried reconstituted milk by using label-free proteomics techniques, respectively. Heat treatment resulted in a significant decrease in the relative intensity of MFGM proteins, such as xanthine dehydrogenase/oxidase, butyrophilin subfamily 1 member A, stomatin, and SEA domain-containing protein, which mainly come from the membrane, while the proteins in skimmed milk, such as β-lactoglobulin, casein, and osteopontin, increased in MFGM after heat treatment. Among these different heat treatment groups, the procedure of spray-drying resulted in the least abundance reduction of caprine milk MFGM proteins. Additionally, it showed heating is the key process affecting the stability of caprine MFGM protein rather than the spray-drying process. These findings provide new insights into the effects of heat treatment on caprine MFGM protein composition and potential biological functions.

## 1. Introduction

The role of milk in the growth and development of newborn mammals is irreplaceable, and it is also an important source of high-quality protein supplements for humans. Although bovine milk occupies a major position in the dairy market, the use of caprine milk in liquid milk, yogurt, and infant formula milk powder has grown rapidly in recent years, with ~20.6 million tons of raw caprine milk produced in 2020 [1]. Additionally, the special nutritional composition of caprine milk provides relatively lower sensitization and higher digestibility for infants and young children in comparison to bovine milk [2,3]. Therefore, increasing attention has been paid to caprine milk.

Similar to bovine milk, protein, fat, and carbohydrates are three main nutritional components in caprine milk. Milk protein, as a key indicator for measuring nutritional value, has been much investigated by scholars. Milk proteins can be divided into three parts: casein, whey protein, and milk fat globule membrane (MFGM) proteins. Casein and whey protein are the two dominant proteins, which account for more than 95% of total milk proteins, whereas MFGM account for 1–4% of total milk proteins [4]. MFGM is composed of a lactone monolayer and a complex lipid bilayer [5], and MFGM proteins are mainly distributed on the inner and outer surfaces of the outer phospholipid bilayer [6]. Although MFGM proteins account for a very small percentage of the total milk proteins, they have gained increasing attention due to their special physiological functions, including antibacterial and anti-cancer properties, and their use for the relief of multiple sclerosis (MS) and autism [7,8,9].

An early investigation of caprine MFGM proteins preliminarily demonstrated their uniqueness by using 1D electrophoresis, and it was found that lactadherin (MFGE8) was the main differential MFGM protein compared to that of bovine milk [10,11]. With the development of proteomics technology, our understanding of MFGM proteins has been improved significantly as proteomics techniques can identify thousands of proteins in one run, especially for relatively low-abundance proteins [12]. For instance, proteomics techniques have been used in studying the difference in the MFGM proteins of colostrum and mature milk, as well as those of different breeds. Lu et al. and Sun et al. studied the differences in MFGM proteins of Guanzhong caprine and Xinong Saanen caprine colostrum and mature milk, and found that colostrum contains more acute phase proteins than mature milk [13,14].

However, MFGM proteins not only vary during physiological processes but also change during processing. In the modern dairy industry, heat treatment is an indispensable technique by which to control the level of microorganisms and ensure the safety of dairy products. Generally, the shelf life of milk after pasteurization is about 2 weeks; in contrast, UHT technology can extend the shelf life of milk to 6 months or more [15]. Ultra-pasteurization is used in the production of cheese and yogurt to ensure the quality of the product, especially in terms of texture and rheology [16,17]. Spray-drying technology is a processing technology that uses milk as the main raw material to produce milk powder. The granular product formed by dehydration of milk in a high-temperature vacuum environment can be stored for 2 years [18]. Although milk is a thermally stable system, milk proteins still have physical and chemical changes during heat treatment. MFGM proteins also change significantly during thermal processing [6,19], which may affect processing performance and influence the nutritional function of dairy products. Proteins such as β-lactoglobulin and α-lactalbumin, which originally existed in whey, were bound to MFGM through disulfide bonds and hydrophobic interactions after heat treatment [20,21]. Furthermore, heat treatment may also affect the iron chelation of milk fat globular membrane proteins [22]. In addition, heat treatment resulted in the lactosylation of MFGM proteins, which could affect the nutritional value, biological function, and safety issues [23]. At present, most previous studies have focused on the effect of heat treatment on caprine whey protein and casein [24,25]. However, few studies have been performed on the effect of heat treatment on caprine MFGM proteins. A previous study showed that heating leads to a loss of caprine MFGM proteins, while under the same conditions (65 °C, 30 min), the heat sensitivity of caprine MFGM proteins is higher than that of bovine MFGM proteins [12]. Caprine milk has smaller-sized fat globules compared to those in bovine milk [12], which leads to different interaction areas of proteins in the milk fat globule membrane and skim milk during the heating process. The low stability of casein micelles in caprine milk relative to bovine milk may also influence the changes in caprine MFGM proteins under different heating environments [26]. At the same time, changes in bovine origin MFGM proteins during heat treatment such as pasteurization and ultra-high temperature sterilization have been reported clearly, especially changes in many low-abundance proteins [6]. Moreover, MFGM proteins of bovine origin have been used in infant formula powders [27]. In contrast, there are limited studies on the changes in caprine MFGM proteins during heat treatment. In our previous research, we analyzed the effect of conventional pasteurization on caprine MFGM proteins, but conventional pasteurization cannot cover a large number of application scenarios [12]. As the stability of MFGM proteins is related to the intensity of heat treatment [20], the changes in caprine MFGM proteins under different heating intensities have not been reported yet.

Therefore, this study aimed to investigate the changes in caprine MFGM proteins during higher-intensity heat treatment by using label-free proteomics techniques, which may provide scientific advice for the better utilization of caprine milk resources.

## 2. Materials and Methods

### 2.1. Sample Treatment and Reagents

Raw milk (RAW): Untreated fresh Saanen caprine milk from 45 lactating Saanen goats (lactating days > 15 d) was mixed and collected in refrigerated storage tanks at 4 °C using an automatic milk extractor at 6:00 a.m. from the ranch of Hangzhou Yunquan Yue Animal Husbandry Co., Ltd. (Hangzhou, China). The samples were transported to the laboratory using a refrigerated truck at 4 °C within 5 h and stored at 4 °C for subsequent analysis. Ultra-pasteurized milk (UP): We took an appropriate RAW sample amount and placed it in a constant temperature water bath (85 ± 0.5 °C) where it was kept for 30 min and stored at 4 °C for subsequent analysis. Ultra-high temperature instant sterilization milk (UHT): We took an appropriate amount of RAW and used 135 °C ± 0.4 °C heat for 5 s, bottled and sealed it, and stored it at 1–3 °C for subsequent analysis. Spray-dried caprine milk powder reconstituted milk (SP): We took an appropriate amount of RAW and placed it in the spray-dryer at an air inlet temperature of 160 °C and an air outlet temperature of 80 °C ± 3 °C. After spray-drying, the processed milk powder was sealed in a sealed bag and stored at 4 °C. We took 1 g of spray-dried sample and fully dissolved it in 9 ml of water to make SP and stored it at 4 °C for subsequent analysis.

Potassium dihydrogen phosphate, sodium dihydrogen phosphate, potassium chloride, sodium hydroxide, calcium chloride, sodium chloride, ammonium persulfate, Coomassie Brilliant Blue R250, acetone, glycerin, methanol, ethanol, glacial acetic acid, and other reagents were purchased from Sinopharm Group. Acrylamide, methylene acrylamide, tris, sodium dodecyl sulfate (SDS), N,N,N′,N′-tetramethylethylenediamine (TEMED), glycine, and imidazole were purchased from Sangon Biotech (Shanghai, China). Two sets of sample staining solution and β-mercaptoethanol were purchased from Bio-Rad. The BCA protein quantitative analysis kit was purchased from Thermo Fisher Scientific (San Jose, CA, USA). All the reagents used in the experiment were of analytical grade (AR), and the experimental water was ultrapure water.

### 2.2. MFGM Protein Extraction

The extraction method for MFGM proteins was based on previous studies [12]. Different groups of caprine milk samples were centrifuged at 1500× *g* for 15 min at 10 °C. The fat of the centrifuged sample was transferred to a new test tube, and washed with 0.1 M PBS buffer at a ratio of 1:1 (v:v), and then centrifuged at 1500× *g* for 15 min at 10 °C. We repeated the washing step four times. Finally, the washed milk fat was diluted to a ratio of 1:1 (v:v) with 0.4% SDS, sonicated for 1 min, and centrifuged at 4 °C and 1500× *g* for 15 min. The fat layer was removed, and the MFGM proteins were in the water phase. We used the BCA kit method to determine the protein concentration. Preparation of the protein samples was performed on the day during which the different heat treatments were completed, and then they were stored at −80 °C until testing. Each set of samples was repeated twice.

### 2.3. Sodium Dodecyl Sulfate–Polyacrylamide Gel Electrophoresis (SDS-PAGE)

We configured 12% separation gel and 4% concentrated gel. We used the sample buffer to divide the caprine milk MFGM protein samples extracted and quantified in the protein sample preparation into a reduced group and a non-reduced group, and adjusted the concentration to 1 mg/mL with 0.4% SDS. To these, β-mercaptoethanol was added to the reduced group to a final concentration of 10%, and the non-reduced group did not have the addition of β-mercaptoethanol. After mixing, the samples were placed in a boiling water bath for 2 min, cooled to room temperature, and we dripped the sample staining solution. After shaking, we took 20 μL and loaded each sample. The initial setting voltage was constant at 60 V. When the sample strip moved to the junction of the concentrated gel and the separating gel, the voltage was constant at 120 V. After the electrophoresis was completed, we moved the electrophoresis adhesive paper in Coomassie Brilliant Blue Staining Solution and shook it for 6 h at room temperature at 25 °C. Then, we poured the dyeing solution and shook it until the background of the rubber sheet became colorless and transparent.

### 2.4. Protein Digestion

We took 100 μL of caprine milk MFGM protein samples of each group in a 1.5 mL centrifuge tube, and diluted the SDS concentration to less than 0.1% with 0.5 M TEAB. Trypsin enzyme is added to the enzyme hydrolysate at a ratio of 1:20 (m:m) to the substrate protein, shaken well and mixed, vortexed, centrifuged at a low speed at 400× *g* for 1 min, and incubated at 37 °C for 4 h. After the digestion was completed, a desalting treatment was performed, and the obtained peptides were freeze-dried after the desalting.

### 2.5. Liquid Chromatography-Tandem Mass Spectrometry Analysis (LC-MS/MS)

The dried peptide sample was reconstituted with mobile phase A (2% ACN and 0.1% FA), centrifuged at 20,000 g for 10 min, and the supernatant was taken for injection. The separation was performed using a Thermo UltiMate 3000 UHPLC. The sample first entered the trap column for enrichment and desalination, and then it was connected in a series with a self-packed C18 column (75 μm inner diameter, 3 μm column particle size, and 25 cm column length), and was separated by the following effective gradient at a flow rate of 300 mL/min: 0–6 min, 6% mobile phase B (98% ACN and 0.1% FA); 6–40 min, mobile phase B, linearly rising from 6% to 25%; 40–48 min, mobile phase B, rising from 25% to 40%; 48–51 min, mobile phase B, increasing from 40% to 90%; 51–55 min, 90% mobile phase B; and 55.5–60 min, 6% mobile phase B. The end of the nanoliter liquid phase separation was directly connected to the mass spectrometer.

The peptides separated in the liquid phase were ionized by the nanoESI source and then entered the tandem mass spectrometer Q-Exactive HF X (Thermo Fisher Scientific, San Jose, CA, USA) for DDA (data-dependent acquisition) mode detection. The main parameter settings were as follows: the ion source voltage was set to 1.6 kV; the scanning range of the primary mass spectrum was 350~1600 *m*/*z*, and the resolution was set to 70,000; the initial *m*/*z* of the secondary mass spectrum was fixed at 100, and the resolution was 17,500. The screening conditions for the precursor ions of the secondary fragmentation were as follows: a charge of 2+ to 7+ and a peak intensity of more than 10,000 were ranked in the top 20 precursor ions. The ion fragmentation mode was HCD, and fragmented ions were detected in Orbitrap. The dynamic rejection time was set to 15 s. The AGC was set to Level 1 3 × 10^6^ and Level 2 1 × 10^5^.

### 2.6. Data Analysis

We used MaxQuant (http://www.maxquant.org/, accessed on 4 February 2022) to identify and quantify the mass spectrometry data of this experiment. The software version used was MaxQuant 1.5.3.30. During operation, we used the original off-machine data as the input file, configured the corresponding parameters and the database, and then performed identification and quantitative analyses. The parameters were selected as follows: trypsin enzyme, minimal peptide length of 7; a PSM-level FDR and protein-level FDR of 0.01; the fixed modification used cabamidomethyl; and the database used uniprot_capra_hircus_nr.fasta (35,280 sequences).

The analysis of MFGM was performed qualitatively and quantitatively using Perseus 1.6.15 software on screened data based on the database. The identified MFGM proteins were characterized by a Venn diagram using the following online website: http://bioinformatics.psb.ugent.be/webtools/Venn/ (accessed on 6 March 2022). For the bioinformatics analysis, the four groups of the identified MFGM proteins were annotated with gene ontology (GO) by a cluster profiler using the protein annotations downloaded from uniport. The pathway analysis was performed using the Kyoto Encyclopedia of Genes (KEGG, https://www.kegg.jp/, (accessed on 14 June 2022)) for significantly different MFGM proteins as compared to the heat treatment groups with RAW. In addition, the images were combined using Photoshop 2018.

## 3. Results

### 3.1. SDS-PAGE of MFGM Proteins

The number and intensity of the bands were greater in the reduced condition than those in the non-reduced condition (Figure 1A,B). This phenomenon had a different manifestation in the heat treatment groups, among which RAW and UHT had the most bands (lines 3, 4, and 5). The MFGM proteins changed differently under different heat treatment conditions. The MFGM proteins were significantly reduced in UP and UHT, mainly in xanthine dehydrogenase/oxidase (XDH), periodic acid Schiff glycoprotein 6/7 (PAS III/IV), butyrophilin subfamily 1 member A1 (BTN1A1), and lactadherin (MFGE8). On the other hand, the casein region and β-lactoglobulin were increased to different extents after heat treatment. Among these, β-lactoglobulin had the highest content in UHT, followed by RAW.

### 3.2. Identification and Quantification of MFGM Proteins

The results presented by the DDA allowed 1326 MFGM proteins to be identified and quantified (Appendix A) from the different heat treatment groups, of which there were 1015 proteins in RAW, 738 proteins in SP, 637 proteins in UP, and 508 proteins in UHT, as shown in Figure 2. Approximately one-third of the common MFGM proteins were identified in RAW and the different heat treatment groups. Among all the identified proteins, unique proteins indicated that 249 proteins were uniquely present in the RAW, as well as 47, 66, and 49 MFGM proteins being uniquely present in SP, UP, and UHT, respectively, as compared to RAW.

### 3.3. Correlation and PCA Analysis of MFGM Proteins from Different Heat Treatment Groups

The Pearson correlation scores (R value) of the two replicates for each group of MFGM protein analysis were above 0.98 (Figure 3A). This showed that the label-free proteomics technique used in this experiment was reproducible. The components of the MFGM proteins in RAW and SP were the most similar to each other, with R values above 0.75 (Figure 3A). The PCA profiles showed significant differences between the different heat treatment groups, while PC1 to PC2 could explain 52.3% to 21.1% of the variance of the MFGM protein components from the different heat treatment groups (Figure 3B). RAW and SP were relatively close. Additionally, they were distributed on the right side of the coordinate system together. UP and UHT were on the left side of the coordinate system, separated along the PC2 score.

### 3.4. Significant Differences in the Changes MFGM Proteins among Different Heat Treatment

In addition to unique proteins, there were 378 common MFGM proteins, as shown in the hierarchical cluster (Figure 4A). These common proteins were divided into two main clusters based on heat treatment conditions. The hierarchical cluster analysis showed that there were significant differences in the changes of MFGM proteins among heat treatment groups, which was consistent with the PCA analysis results.

We further compared the changes in the different heat treatment groups using Student’s T test with Benjamini–Hochberg FDR ≤ 0.05 (−Log *p*-value > 1.3010) and |Log_2_ Fold Difference|>2 as the cutoff for the significant differences of the proteins between the two groups. Proteins were found to be significant upregulated and downregulated in the different heat treatment groups as compared to RAW. We used volcano maps to visualize these significant differences (Figure 4B–D). Obviously, most of the common proteins were decreased in the heat treatment groups. From the perspective of the amount and dispersion of downregulated proteins, the decrease in common MFGM proteins in UP and UHT was significantly greater than that of SP. There were 192, 185, and 37 down-regulated proteins, respectively. Additionally, there were 23, 10, and 17 significantly upregulated proteins in the UP, UHT, and SP, respectively, as compared with the RAW.

### 3.5. GO Analysis of Caprine MFGM Proteins in Relation to Different Heat Treatments

We performed gene ontology annotation analysis according to the three categories of biological processes, cellular components, and molecular functions on all the MFGM proteins from the different heat treatment groups (Figure 5). As shown in Figure 5, in terms of molecular functions, binding, catalytic activity, and structural molecule activity were the top three in RAW and UP, while binding, catalytic activity, and molecular function regulation were the most important molecular functions in SP and UHT. In terms of cellular components, apart from the cell, cell part, and organelle sources of the same genus, more of RAW is derived from the membrane, while the other heat treatment groups were derived from the extracellular region. In terms of biological processes, MFGM proteins are mainly involved in cellular processes and biological regulation, but they were more involved in metabolic processes in RAW and more involved in biological process regulation in the heat treatment groups.

The unique proteins were caprine MFGM proteins found in UP (66), SP (47), and UHT (49) as compared with RAW. A total of 249 unique proteins were found in RAW as compared with the other heat treatment groups. We also performed GO analysis on these unique proteins. As shown in Appendix A, there were more unique MFGM proteins of the different heat treatment groups enriched in the extracellular region of the cellular component.

### 3.6. Pathway Analysis of Unique MFGM Proteins and Significantly Different MFGM Proteins

The first 20 KEGG pathway analyses were divided into six primary levels: cellular processes, genetic information processing, human diseases, metabolism, and organismal systems of significantly different MFGM proteins between the different heat-treated groups, including downregulated and upregulated proteins as compared with those in RAW (Figure 6) and unique proteins (Appendix A).

The same KEGG pathway analysis was also used to identify significantly different MFGM proteins, as shown in Figure 6. In terms of significantly downregulated proteins, most of the proteins were enriched in the translation of genetic information processing in SP vs. RAW; most of proteins were enriched in signal transduction of environmental information processing in UP vs. RAW; and most of proteins were enriched in immune system functions in UHT vs. RAW. On the other hand, significantly upregulated MFGM proteins in the heat treatment groups were mainly enriched in human diseases and organismal system pathways.

As shown in Appendix A, 248 unique MFGM proteins of RAW were mainly derived from global and overview maps of metabolism, folding, sorting, and degradation in genetic information processing and neurodegenerative diseases in human diseases. As shown in Appendix A–D, there were 47, 66, and 49 unique MFGM proteins of SP, UP, and UHT, respectively, and they were more enriched in human diseases and the immune system in the physiological system.

## 4. Discussion

### 4.1. Reliability of the Identified MFGM Proteins by Using Label-Free Proteomics Technique

In this experiment, a total of 1119 caprine MFGM proteins were identified and quantified using label-free proteomics technology; among them, RAW had 1015 identified MFGM proteins from the mixed Saanen caprine milk samples (more than 30), which were higher than previous studies. In a recent study, 734 MFGM proteins were identified in Saanen caprine colostrum and mature milk [13]. Using the same techniques, Sun identified 593 MFGM proteins in Guanzhong caprine milk [28]. Another study also identified 423 MFGM proteins using mixed samples of five Guanzhong caprine milk samples from colostrum and mature milk [29].

The relatively higher number of caprine MFGM proteins found in our study can be attributed to the following reasons: (1) The number of samples. In Lu’s study, the milk samples were collected from five goats, while 30 were used for collection in Sun’s studies. In this study, 45 caprine milk samples collected on-site were used. The rich source of caprine milk brings a wealth of MFGM protein types due to individual differences [30]. (2) Lactation stage of samples. The mixture of colostrum and mature milk may result in a high number of identified proteins. In the research on caprine colostrum and mature milk, unique proteins account for approximately one-third of the MFGM protein components [13,29]. The samples used in this study were mixed with caprine milk from different lactation periods, which may result in more abundant MFGM protein components. (3) Advanced proteomics technology. The Q-Exactive HF X has better resolution and detection limits than the previous Q-Exactive Plus and LTQ, which may have led to the higher number of identified proteins in our study compared to the other previous studies.

In addition, the Pearson correlation coefficient between the two technical replicates was above 0.98 (Figure 3), indicating that this experimental proteomics technology has stable reproducibility. On the other hand, the mass spectrometry results also match the electrophoresis results. These ensure the accuracy of the label-free qualitative and quantitative comparison.

### 4.2. Changes in the Caprine MFGM Proteins after Different Heat Treatment

After heat treatment, a certain number of proteins decreased or were unidentified, as shown in both the SDS-PAGE (Figure 1A) and proteomic results, which were similar to those of a previous study [12,31]. This decrease and non-identification may be due to changes in the structure and state of the proteins, as follows. Firstly, the MFGM proteins were transferred to the liquid phase of the milk due to the temperature, such as PAS6/7, XDH, and STOM (stomatin) [12,32]. Secondly, heat treatment can modify the MGFM proteins, which may cause the proteins to resist enzymatic digestion or/and cause the peptide to have new modifications after digestion, such as glycosylation of lysine [6,33]. In general, the amount of caprine MFGM protein was reduced after heat treatment.

Furthermore, the disappearance and decrease in MFGM proteins were related to heating intensity, and compared with RAW more MFGM proteins were significantly down-regulated in UHT (Figure 4D) than in SP and UP (Figure 4B,C), which is consistent with previous studies. In our study, more than 25% of the MFGM proteins were identified in UP in comparison to than UHT, suggesting that UHT had a greater impact on caprine MFGM proteins than UP in terms of the types of MFGM proteins. Additionally, the decrease in some common MFGM proteins in UP was greater than that of UHT (Appendix A). Previous studies have also shown that bovine MFGM proteins at 45–60 KD were significantly reduced in UHT in comparison to pasteurized milk [23]. For instance, membrane glycoproteins, such as the PAS domain, were found to be reduced after heat treatment due to whey protein (mainly β-lactoglobulin) replaced it on the fat globule membrane [32]. In sum, the longer duration of ultra-pasteurization and UHT have a significantly higher impact on MFGM than traditional pasteurization.

On the other hand, spray-drying had a less noticeable effect on the MFGM protein number and abundance as compared with the other heat treatments in this study, which differs from the results of previous reports. In a previous study, the authors found the content of MFGM proteins in commercial milk powder to be much lower as compared to RAW [23]. We believe this difference is due to the range of applied processing before spray-drying, such as preheating and concentration during the manufacture of whole milk powder [25,34]. We did not perform any preheating before spray-drying in our study, but infant formula has to undergo preheating and concentration before spray-drying. On the other hand, although the inlet temperature of the spray-drying barrel can be as high as 160 °C, a sample that is separated into small emulsion droplets in a vacuum environment will be dehydrated quickly, and the actual wet bulb temperature of the emulsion droplet sample will be lower. Therefore, preheating had a much stronger influence on the MFGM proteins than that of spray-drying.

### 4.3. Potential Physiological Functions Changes in MFGM Proteins after Different Heat Treatments

The molecular function of unique proteins in RAW was mainly related to binding, catalytic activity, structural molecule activity, and molecular function regulation, suggesting that these disappeared or decreased proteins in the heat-treated sample may have lost some physiological functions for humans (Figure 5). For instance, lysozyme was only identified in RAW, and it has a primarily bacteriolytic function. Those in tissues and body fluids are associated with the monocyte–macrophage system and enhance the activity of immune agents [35,36]. Another MFGM protein, BTN1A1, decreased after heat treatment. BTN1A1 is a major protein in milk fat droplets; it belongs to the immunoglobulin superfamily and may inhibit the proliferation of CD4 and CD8 T-cells activated by anti-CD3 antibodies, T-cell metabolism, and IL2 and IFNG secretion [37]. XDH is a key enzyme in purine degradation, which can catalyze the oxidation of hypoxanthine to xanthine and catalyze the oxidation of xanthine to uric acid. Its change during heat treatment was found to be the same as that in BTN1A1, and this trend is consistent with previous studies [12,25]. C3 plays a central role in the activation of the complement system and has been widely reported to play a protective role when milk is used to feed newborns [38,39]; it is also reduced to varying degrees during the heat treatment process (Figure 4).

The KEGG annotation analysis of significantly different proteins can help us better understand the effect of heat treatment on the function changes in caprine MFGM proteins. Significantly reduced proteins in UP and UHT with a higher heat intensity were involved in the immune system pathways of the organic system and human diseases (Figure 6C,E), which is also related to the membrane in cellular components such as endoplasmin (HSP90B1) Marcks-related protein (MARCK), and small monomeric GTPase (RALB). UP and UHT have significant effects on the immune system-related MFGM proteins, but the effect is relatively small in the SP, which is consistent with previous conclusions. Regarding the KEGG analysis of the significantly upregulated proteins, many of them were related to immunity (Figure 6B,D,F). For instance, the S100-A8 (S100A8) and S100-A9 (S100A9) proteins were related to innate immune response in the IL-17 signaling pathway of the immune system. We speculated that these skim milk proteins combined with MFGM proteins through disulfide bonds or van der Waals forces during thermal processing. This may be one of the reasons why immune functional proteins were reduced in skim milk after heat treatment [40,41].

In general, these results reflect the changes in MFGM proteins composition which lead to potential functional changes in caprine milk MFGM proteins after heat processing. Additionally, a previous study reported that lactosylated lysine cannot be recognized by gastrointestinal proteases [42]. In addition, more mechanisms of nutrient function changes caused by heat treatment need to be confirmed in further studies.

### 4.4. The Interaction between Skimmed Milk Proteins and MFGM Proteins under Heat Treatment

Comparing the reduced and non-reduced MFGM protein images in SDS-PAGE, it was clearly seen that some of the MFGM proteins bind to the proteins in the milk emulsion. Therefore, the bands were not obvious in the SDS-PAGE image in the reduced state. However, the difference can be observed in the proteomic results (Appendix A). In the MFGM fraction, binding occurred through disulfide bonds including β-lactoglobulin and α-lactalbumin with XDH/XO and BTN [43]. This situation was also consistent with the results of our previous research [12]. Previous studies have shown that milk protein β-lactoglobulin and casein are combined to MFGM as the heating intensity increases [44], and the combination is gradually strengthened from 65 °C to 85 °C [41]. Until recently, researchers believed that the main reason is that the free sulfur groups of MFGM were changed to disulfide bonds with β-lactoglobulin after heating, which leads to the combination of MFGM proteins and β-lactoglobulin [21,32]. In addition, the casein micelles adsorbed by the milk fat globules will react with skimmed milk proteins during the heat treatment process, and β-lactoglobulin will combine with the casein micelles to enter the MFGM component [20]. However, in more recent research, non-covalent bonding, such as hydrogen bonding, electrostatic and hydrophobic interactions are also important links in the binding of whey proteins and MFGM proteins [45].

The interaction of skim milk proteins and MFGM can partially explain the decrease in skim milk proteins after heat treatment. However, the effect of this combination on the function of skimmed milk proteins is temporarily unclear, and further studies are needed to confirm the impact of this nutritional function change.

## 5. Conclusions

On the whole, heat treatment can cause a reduction in MFGM protein components, and different proteins have different sensitivities to heat treatment; this is related to the actual heating temperature and time. In this study, spray-drying had the least effect on the changes of MFGM proteins, in that it resulted in more kinds of remaining MFGM proteins, especially many low-abundance MFGM proteins as compared with ultra-pasteurization and ultra-high-temperature instant sterilization treatment. The results in this study showed that the heating is the key process affecting the stability of caprine MFGM protein rather than the spray-drying process. This experiment provides new insights into the functional changes that may be caused by heat treatment of caprine MFGM proteins, especially in the immune system, and provides necessary support for the further application of caprine milk resources.

## Figures and Tables

**Figure 1 foods-11-02705-f001:**
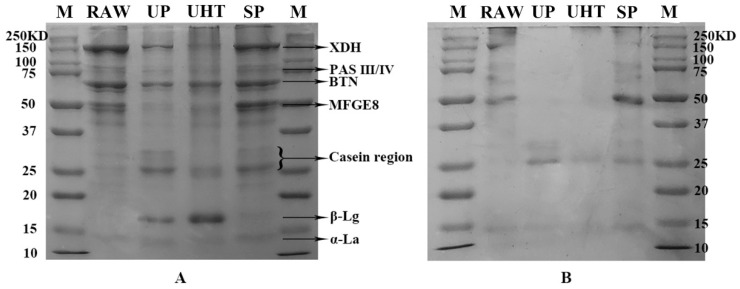
SDS-PAGE images of MFGM proteins from RAW, untreated milk; UP, ultra-pasteurized milk; UHT, ultra-high temperature instant sterilization milk; SP, spray-dried milk with reduced conditions (**A**) and non-reduced conditions (**B**). Line M is the molecular weight standard.

**Figure 2 foods-11-02705-f002:**
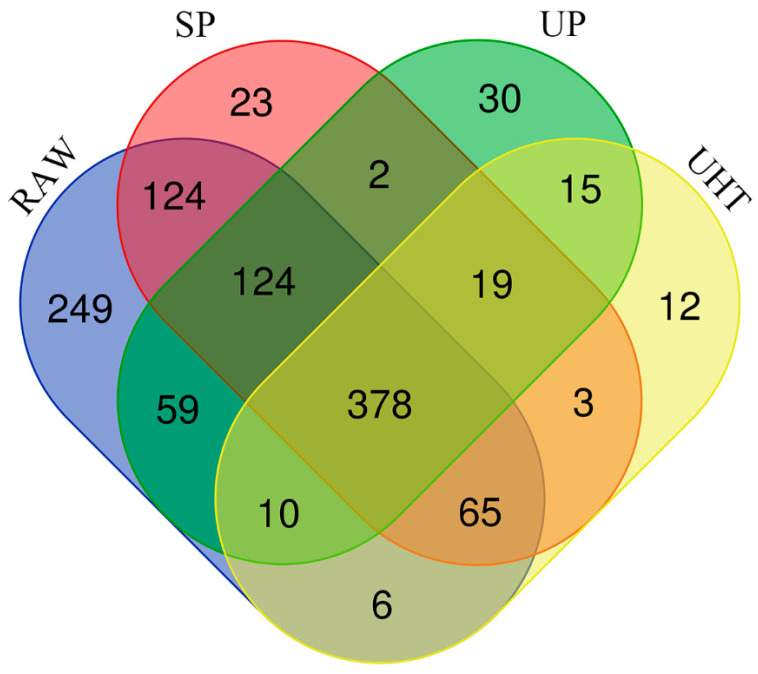
Venn diagram of the identified and quantified MFGM protein components from RAW, untreated milk; UP, ultra-pasteurized milk; UHT, ultra-high temperature instant sterilization milk; SP, spray-dried milk.

**Figure 3 foods-11-02705-f003:**
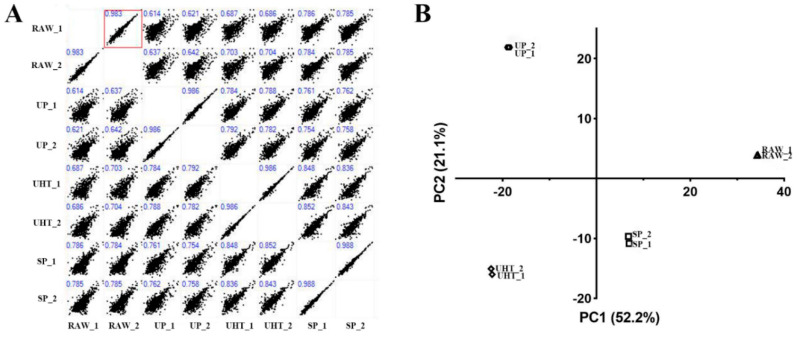
Pearson correlation score diagram (**A**) showing the correlation of caprine MFGM proteins between the different experimental groups. The score in the upper left corner of each small scatter plot is the R value of the experimental groups on the corresponding abscissa and ordinate. PCA score plot (**B**) showing the identified and quantified caprine MFGM proteins of the different heat treatment groups.

**Figure 4 foods-11-02705-f004:**
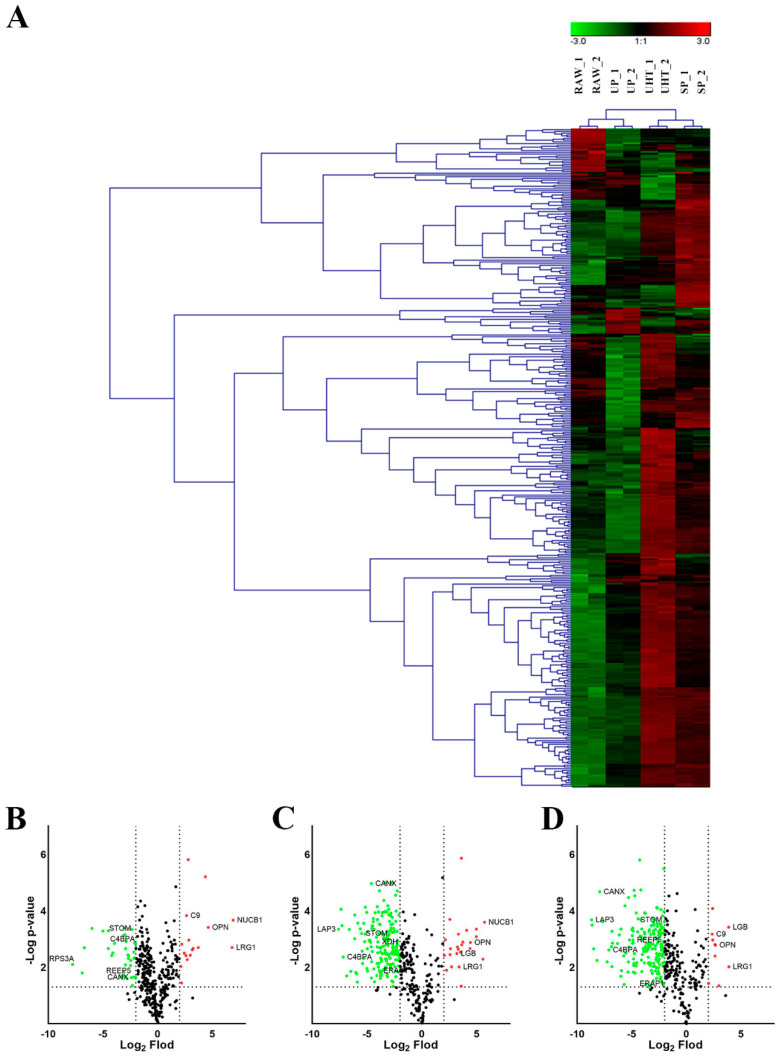
Hierarchical clustering chart (**A**) showing the cluster of the common MFGM proteins from RAW, untreated milk; UP, ultra-pasteurized milk; UHT, ultra-high temperature instant sterilization milk; SP, spray-dried milk. The numbers at the end represent different repetitions. Volcano plot showing the significant differences of MFGM proteins in SP vs. RAW (**B**), UP vs. RAM (**C**), and UHT vs. RAW (**D**). Green dots indicate significantly downregulated proteins; red dots indicate significantly upregulated proteins.

**Figure 5 foods-11-02705-f005:**
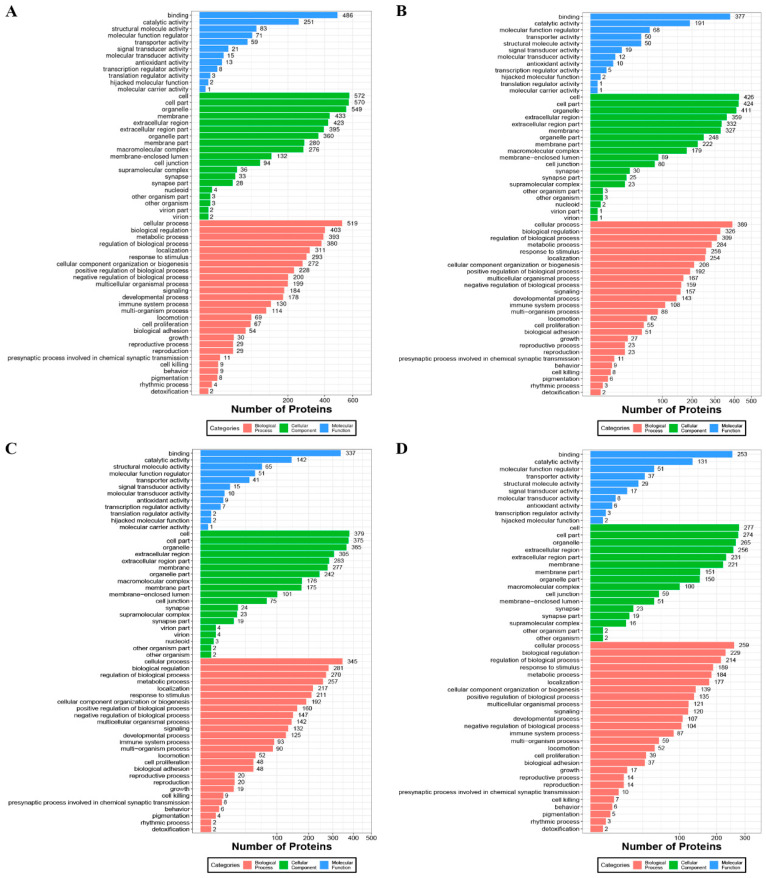
GO analysis of the identified MFGM proteins of raw milk (**A**), spray-dried milk (**B**), ultra-high temperature instant sterilization milk (**C**), and ultra-pasteurized milk (**D**), respectively.

**Figure 6 foods-11-02705-f006:**
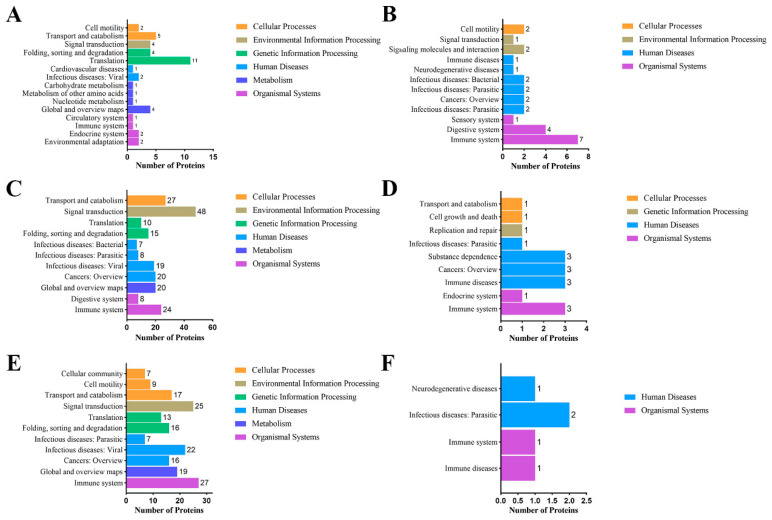
KEGG pathway analysis of significantly different MFGM proteins comparing the heat treatment groups with RAW. (**A**,**C**,**E**) are significantly downregulated MFGM proteins, and (**B**,**D**,**F**) are significantly upregulated proteins of SP vs. RAW, UP vs. RAW, and UHT vs. RAW, respectively.

## Data Availability

Data is contained within the article and Appendix A.

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
