# Peer review of "Changes in Caprine Milk Fat Globule Membrane Proteins after Heat Treatment Using a Label-Free Proteomics Technique"

_foods, 2022, doi:10.3390/foods11172705_

Round 1
Reviewer 1 Report
The authors have studied the changes of caprine MFGM protein by three heat-treatment processes of ultra-pasteurization (85°C, 30min), ultra-high temperature instant sterilization (135°C, 5s) and spray drying (inlet 160℃, outlet 80℃) through label-free proteomics technique. To sum up, the manuscript is well written, and the results were well discussed. Besides, the study presented novel results and beneficial for dairy industry.
Introduction:
Please provide the properties of caprine MFGM protein as affected by different processes compared to protein from other sources as mentioned in the literature review.
Conclusion:
Please provide the future studies related to the caprine MFGM protein.

Author Response
# Reviewer1
Introduction:
Please provide the properties of caprine MFGM protein as affected by different processes compared to protein from other sources as mentioned in the literature review.
AU: Thanks for your comments. We have added the changes of caprine milk MFGM proteins during heat-treatment compared to other species in the introduction.

Reviewer 2 Report
The manuscript includes the original data on the effect of ultra-pasteurization, ultra-high temperature instant sterilization, and spray drying on the changes in caprine MFGM proteins.
The results are very interesting and can be useful in practice. However, some issues need to be improved.
All abbreviations should be explained when used for the first time. The abstract contains a lot of abbreviations without explanation.
Please provide information on the current situation and newer references for the sentence: 'Although bovine milk occupies a major position in the dairy market, caprine milk has grown rapidly in liquid milk, yogurt and infant formula milk powder in recent years 1.'
It has been written: 'Few studies were performed on the effect of heat-treatment on goat milk MFGM.': What do few studies mean? Please provide more details for this sentence.
45 samples of mixed Saanen caprine milk without any heat treatment were collected. Please provide more details on the collection of samples.
The number of repetitions should be specified for each type of measurement.
Section 2. Materials and methods is difficult to read. Please pay attention to the language and the style. Other sections also require extensive editing of English language.
Subsection 2.6. Data analysis doesn't include all the analyzes, the results of which are included in section 3. Results. Subsection 2.6. should be more detailed.
It has been written: "The Pearson correlation score (R value) of two replicates for each group MFGM protein analysis were above 0.98". How many cases were included in each replicate?
The conclusions should be more comprehensive and supported by the results.
Author Response
# Reviewer2
The manuscript includes the original data on the effect of ultra-pasteurization, ultra-high temperature instant sterilization, and spray drying on the changes in caprine MFGM proteins.
AU: Thanks for your comments. Raw data has been uploaded to the system as supplementary files.
The results are very interesting and can be useful in practice. However, some issues need to be improved.
All abbreviations should be explained when used for the first time. The abstract contains a lot of abbreviations without explanation.
AU: Thanks for your comments. The full name of abbreviations that appear for the first time have been added in the revised manuscript.
Please provide information on the current situation and newer references for the sentence: 'Although bovine milk occupies a major position in the dairy market, caprine milk has grown rapidly in liquid milk, yogurt and infant formula milk powder in recent years 1.'
AU: Thanks for your comments. Newer reference has been revised, while detailed caprine milk production have been added to the introduction. (with ∼20.6 million tons of raw caprine milk produced in 2020)
It has been written: 'Few studies were performed on the effect of heat-treatment on goat milk MFGM.': What do few studies mean? Please provide more details for this sentence.
AU: Thanks for your comments. The details have been added as shown below.
A previous study showed that the heat sensitivity of caprine MFGM protein was higher than that of bovine MFGM protein under the same conditions (65°C, 30min). Caprine milk has smaller size fat globules compared to bovine milk, which leads to different interaction areas of proteins in the milk fat globule membrane and skim milk during the heating process. The low stability of casein micelles in caprine milk relative to bovine milk may also influence the changes in caprine MFGM protein under different heating environments.
45 samples of mixed Saanen caprine milk without any heat treatment were collected. Please provide more details on the collection of samples.
AU: Thanks for your comments. We have provide the details of sample collection in the revised manuscript.
‘Fresh Saanen caprine milk from 45 lactating Saanen caprine was mixed and collected in refrigerated storage tanks at 4°C using an automatic milk extractor at 6:00 am from the ranch of Hangzhou Yunquan Yue Animal Husbandry Co., Ltd. Transported to the laboratory using a refrigerated truck at 4℃ within 5h and stored at 4℃ for subsequent analysis.’ The details have been added in section 2.1 after ‘Raw milk (RM)’.
The number of repetitions should be specified for each type of measurement.
AU: Thanks for your comments. Two replicates of each MFGM proteome assay were performed for different heat treatment groups. This has been added in the method section.
Section 2. Materials and methods is difficult to read. Please pay attention to the language and the style. Other sections also require extensive editing of English language.
AU: Thanks for your comments. We have revised the English language by using MDPI English editing service with ID “English-48893”.
Subsection 2.6. Data analysis doesn't include all the analyzes, the results of which are included in section 3. Results. Subsection 2.6. should be more detailed.
AU: Thanks for your comments. In subsection 2.6 more details of the analysis of all the analyzes have been added.
It has been written: "The Pearson correlation score (R value) of two replicates for each group MFGM protein analysis were above 0.98". How many cases were included in each replicate?
AU: Thanks for your comments. Each replicate contains a mix of 45cases.
The conclusions should be more comprehensive and supported by the results.
AU: Thanks for your comments. We have modified the conclusion as shown in the revised manuscript.

Round 2
Reviewer 2 Report
The manuscript has been improved
Author Response
Thanks.